# Overview of the Manufacturing Methods of Solid Dispersion Technology for Improving the Solubility of Poorly Water-Soluble Drugs and Application to Anticancer Drugs

**DOI:** 10.3390/pharmaceutics11030132

**Published:** 2019-03-19

**Authors:** Phuong Tran, Yong-Chul Pyo, Dong-Hyun Kim, Sang-Eun Lee, Jin-Ki Kim, Jeong-Sook Park

**Affiliations:** 1College of Pharmacy, Chungnam National University, 99 Daehak-ro, Yuseong-gu, Daejeon 34134, Korea; phuongtran24288@gmail.com (P.T.); himchani46@naver.com (Y.-C.P.); dong_bal@naver.com (D.-H.K.); nnininn@hanmail.net (S.-E.L.); 2College of Pharmacy and Institute of Pharmaceutical Science and Technology, Hanyang University, 55 Hanyangdaehak-ro, Sangnok-gu, Ansan 15588, Korea

**Keywords:** solid dispersion, classification, manufacturing methods, bioavailability, anticancer drugs

## Abstract

Approximately 40% of new chemical entities (NCEs), including anticancer drugs, have been reported as poorly water-soluble compounds. Anticancer drugs are classified into biologic drugs (monoclonal antibodies) and small molecule drugs (nonbiologic anticancer drugs) based on effectiveness and safety profile. Biologic drugs are administered by intravenous (IV) injection due to their large molecular weight, while small molecule drugs are preferentially administered by gastrointestinal route. Even though IV injection is the fastest route of administration and ensures complete bioavailability, this route of administration causes patient inconvenience to visit a hospital for anticancer treatments. In addition, IV administration can cause several side effects such as severe hypersensitivity, myelosuppression, neutropenia, and neurotoxicity. Oral administration is the preferred route for drug delivery due to several advantages such as low cost, pain avoidance, and safety. The main problem of NCEs is a limited aqueous solubility, resulting in poor absorption and low bioavailability. Therefore, improving oral bioavailability of poorly water-soluble drugs is a great challenge in the development of pharmaceutical dosage forms. Several methods such as solid dispersion, complexation, lipid-based systems, micronization, nanonization, and co-crystals were developed to improve the solubility of hydrophobic drugs. Recently, solid dispersion is one of the most widely used and successful techniques in formulation development. This review mainly discusses classification, methods for preparation of solid dispersions, and use of solid dispersion for improving solubility of poorly soluble anticancer drugs.

## 1. Introduction

Cancer is one of the leading causes of death worldwide, and treatment remains a great challenge. Currently, there are three major cancer treatment strategies of surgery (performed by a surgical oncologist), chemotherapy (use of anticancer drugs), and radiotherapy (delivered by a radiooncologist) [1]. The objective of any treatment is to kill as many cancer cells as possible and minimize death of normal cells. Patients can receive monotherapy or combination therapy. For example, Hwang et al. [2] reported a combination of photodynamic therapy (PDT) and anti-tumor immunity in cancer therapy. Among the three major therapeutic strategies, surgery has been the first line of treatment for many solid tumors. This strategy involves removal of solid tumors by a surgical oncologist under anesthesia. However, patients have to be hospitalized, the entire tumor cannot always be removed, damage can occur to nearby normal tissues, and complications can arise from surgery. Radiotherapy is focused on the tumor and is designed to kill a large proportion of cancer cells within the tumor. As with surgery, this therapy has disadvantages such as damage to surrounding tissues (e.g., lung, heart) and inconvenience for patients (e.g., in some cases, it must be delivered daily, 5 days per week, for 1–2 months). In addition, radiotherapy causes hair loss. As such, use of anticancer drugs (chemotherapy) is currently preferred for treatment of both localized and metastasized cancers. Chemotherapy can kill many cancer cells throughout the body, eradicate microscopic disease at the edges of tumors that may not be seen by a surgeon, and be used in combination with other therapies. Tumor-targeted delivery and controlled release of drugs are two important strategies for improving therapeutic efficacy and reducing side effects. 

Cancer-targeting strategies for drug delivery include passive and active targeting strategies [3]. Active targeting focuses anticancer drugs to ligands or receptors in the target region. For example, folic acid (FA) has been used as a targeting ligand to the folate receptor in various tumor sites including lung, ovarian, breast, and colon cancers [4]. Vinothini et al. [5] developed a graphene oxide-methyl acrylate-FA/paclitaxel (GO-MA-FA/PTX) nanocarrier for targeted anticancer drug delivery to breast cancer cells, resulting in reduction of 39% of typical cytotoxic effects. Voeikov et al. [6] prepared dioxadet-loaded nanogels using a block copolymer of polyethylene glycol and polymethacrylic acid (PEG-b-PMAA) as a high-loading capacity (>35% *w*/*w*) and high-loading efficiency (>75%) drug delivery system. In another study, trastuzumab, a monoclonal antibody with specific targeting to human epidermal growth factor receptor 2 (HER2) protein, was used in combination with cisplatin for treatment of HER2-overexpressing breast cancer [7]. Passive targeting occurs through interactions with the reticuloendothelial system, allowing for entry into the blood, which is dependent on particle size and surface characteristics. Development of nanoparticle anticancer drugs has improved therapeutic efficacy because the drug can be directly and selectively targeted to cancer cells [8,9,10]. For example, Kirtane et al. [11] developed a polymer-surfactant nanoparticle composed of a sodium alginate core complexed with doxorubicin and the surfactant aerosol OT for stability. The relative BA of the nanoparticle formulation was higher than that of the pure drug. In another study by Valicherla el al. [12], docetaxel nanoparticles were prepared in a self-emulsifying drug delivery system (SEDDS) to enhance BA and anti-tumor activity. The BA of docetaxel-SEDDS was 3.19-fold higher than that of the pure drug. Furthermore, docetaxel-SEDDS showed 25-fold higher cytotoxic activity than the free drug in vitro.

Intravenous (IV) and oral are the two most popular routes of drug administration. Paclitaxel is an anticancer drug used to treat many types of cancers such as breast, ovarian, lung, and pancreatic cancer. Paclitaxel is administered by IV infusion and sold under the marketed name Taxol 30 mg (5 mL), 100 mg (16.7 mL), and 300 mg (50 mL) [13]. Tamoxifen, an anticancer drug used to treat breast cancer, is sold under the brand name Nolvadex (10 mg and 20 mg tablets) and is formulated for oral administration [14]. IV infusion is the best route of administration for most anticancer drugs because this route leads to 100% BA. However, IV administration is associated with several side effects, short duration of effectiveness, and inconvenience due to hospitalization. Taxol is prepared by solubilizing paclitaxel in ethanol: Cremophor EL (1:1, *v*/*v*). This formulation is then diluted 5–20-fold in normal saline, resulting in a final concentration of 0.03–0.60 mg/mL [15]. However, several side effects result from administration of Cremophor EL such as hypersensitivity, nephrotoxicity, and neurotoxicity [16,17,18]. Despite the continuing interest on anticancer drugs in recent years, their use in clinical anticancer therapy is limited due to nonspecific biodistribution, low therapeutic indices, and poor aqueous solubility. As such, oral administration has received increasing attention, leading to increased numbers of anticancer drugs being developed for oral dosing. Oral drugs can be administered at home, do not induce the same discomfort as IV infusion, and the drug concentration can be maintained for long time periods in cancerous cells. Oral dosage forms rely on drug solubility to achieve the desired concentrations in the systemic circulation. Drugs have to dissolve in the gastrointestinal (GI) fluid and then permeate the membrane of the GI tract into the blood to be effective. However approximately 40% of new chemical entities (NCEs), including anticancer drugs, are poorly water-soluble [19,20,21]. Due to poor aqueous solubility, the drug cannot completely absorb in the GI tract, resulting in poor bioavailability (BA) and high intra- and inter-individual pharmacokinetic variability. 

The Biopharmaceutics Classification System (BCS) divides drugs into four groups as follows: Class I (high solubility, high permeability), Class II (low solubility, high permeability), Class III (high solubility, low permeability), and Class IV (low solubility, low permeability) (Figure 1) [22]. A drug substance is considered highly soluble when the highest single therapeutic dose is soluble in 250 mL or less of aqueous media over the pH range of 1.2–6.8 at 37 ± 1 °C. Permeability is evaluated on the basis of the extent of absorption of a drug from human pharmacokinetic studies. Alternatively, in vitro culture methods can also be used to predict drug absorption in humans. A drug is considered highly permeable when the absolute BA is ≥85%. High permeability can also be concluded if ≥85% of the administered dose is recovered in urine as unchanged (parent drug) or as the sum of the parent drug, Phase 1 oxidative, and Phase 2 conjugative metabolites. Among four groups, drugs belonging to Class II and IV exhibit poor aqueous solubility, resulting in poor BA. Therefore, enhancing solubility and BA of poorly water-soluble drugs in BCS Classes II and IV is a significant challenge in the pharmaceutical industry. 

In the clinic, there are many insoluble drugs with small dose administration such as risperidone (0.25–4 mg), lorazepam (0.5–2 mg), diazepam (2–10 mg), and clonazepam (0.5–2 mg), which do not require increased solubility. However, the solubility of these drugs is usually affected by pH due to physicochemical properties resulting in a decrease in the effective treatment. For example, risperidone is indicated for treatment of schizophrenia at a small dose (0.25 mg, 0.5 mg, 1 mg, 2 mg, 3 mg, or 4 mg) in oral administration. It is a weak base that is practically insoluble in water. Its solubility is pH dependent, with high solubility in acidic pH, and decreasing solubility as pH increases (range from >200 mg/mL at pH 2.1 down to 0.29 mg/mL at pH 7.6 and reaches 0.08 mg/mL at pH 8). After oral administration, risperidone is rapidly absorbed, and approximately 80% of drugs will be absorbed in the GI tract, where the solubility significantly drops. Moreover, risperidone is a metabolized drug, in which approximately 70% and 14% of the dose is excreted in urine and feces, respectively. Therefore, enhancing solubility in simulated intestinal pH to ensure higher drug concentrations at the main absorption site and improve BA is a challenge in drug development.

To improve the solubility and BA of poorly water-soluble drugs, several methods have been developed such as solid dispersion (SD) [23,24,25], complexation [26], lipid-based systems [27,28], micronization [29,30], nanonization [31,32,33], and co-crystals [34,35]. Among these, SD is one of the most potent and successful methods. SD is defined as a group of solid products consisting of a hydrophobic drug dispersed in at least one hydrophilic carrier, resulting in enhanced surface area, leading to higher drug solubility and dissolution rate. Improving wettability and dispersibility and reducing aggregation and agglomeration of drug particles result in enhanced drug BA. An SD is typically characterized on the molecular level using Fourier-transform infrared spectroscopy (FTIR), Raman spectroscopy, near-infrared spectroscopy (NIR), and solid-state nuclear magnetic resonance (SSNMR) at the particulate level using powder X-Ray diffraction (PXRD), differential scanning calorimetry (DSC), scanning electron microscopy (SEM), and transmission electron microscopy (TEM) and at the bulk level using density, contact angle, flowability, and Karl Fischer titration [36]. SD can be accomplished by several methods such as solvent evaporation [23], hot-melt extrusion [37], and spray drying [38]. In this review, we mainly discuss classification of drugs, methods for preparation of SD, and use of SD for improving solubility of poorly soluble anticancer drugs.

## 2. Solid Dispersions

An SD is defined as a group of solid products consisting of a hydrophobic drug dispersed in at least one hydrophilic carrier, resulting in increased surface area and, enhanced drug solubility and dissolution rate. They are classified as follows.

### 2.1. Carrier-Based Class of Solid Dispersion

Many carriers are used in SD. These carriers determine the final formulation properties and can be categorized into first, second, and third classes (Figure 2).

#### 2.1.1. First Class of SD

The first study of SD was conducted by Sekiguchi and Obi in 1961 [39]. They studied absorption of a eutectic mixture of sulfathiazole [39] and chloramphenicol [40] compared with that of the original formulations of the same drugs. The results showed that the use of urea as a hydrophilic carrier increased the absorption of sulfathiazole and chloramphenicol in the eutectic mixture compared to that of the conventional formulations, thereby improving BA. Sugars and their derivatives are carriers with high solubility in water and low toxicity. Levy [41] and Kaning [42] used mannitol as a carrier to develop SD as a solid mixture instead of as a eutectic mixture. The formulation using mannitol as a carrier showed higher dissolution compared to the original formulation of the drug. To enhance the dissolution profile of clotrimazole, Madgulkar et al. [43] prepared an SD by a fusion method using various sugars such as d-mannitol, d-fructose, d-dextrose, and d-maltose as carriers at a different weight ratios to the drug. The results showed that a 100% solution of mannitol showed an 806-fold increase in solubility compared to the conventional drug in water. The dissolution profile of clotrimazole SD was improved at 1:3 drug to mannitol ratio. In conclusion, urea and sugars were first used as crystalline carriers for production of SD. These formulations were thermodynamically unstable, resulting in slow drug release.

#### 2.1.2. Second Class of SD

Because of thermodynamic instability of first class SD [44], second class SDs were introduced using amorphous polymeric carriers [45] instead of urea or sugars. The polymeric carriers can be synthetic or natural polymers. Synthetic polymers include povidone (PVP) [46,47,48,49,50], PEG [51,52,53,54], and polymethacrylates [55,56], and natural polymers include hydroxypropylmethylcellulose (HPMC) [57,58,59,60,61], ethyl cellulose [62,63,64], and starch derivatives such as cyclodextrins (CDs) [65]. A study by Franco et al. [66] showed that using PVP as a carrier in ketoprofen SD increased the dissolution rate of ketoprofen 4.2-fold compared to that of the conventional drug. In a study by Dhandapani and El-gied [67], β-CD was used as a carrier in cefixime SD to improve solubility. The dissolution rate of cefixime SD was 6.77-fold higher than that of the pure drug. In another study, an SD of diclofenac sodium was prepared by solvent evaporation using Eudragit E 100 as the carrier [68]. Solubility of diclofenac sodium from the SD (0.823 mg/mL) was approximately 58.8-fold higher compared with the pure drug (0.014 mg/mL). In dissolution studies, diclofenac sodium released from SD was approximately 60% after 2 h at pH 1.2, while the pure drug release was less than 10% after 2 h. Second class SDs are dispersed in polymeric carriers and achieve a supersaturated state. These formulations have smaller particle sizes and enhanced wettability thereby increasing the aqueous solubility of drugs.

#### 2.1.3. Third Class of SD

The third class of SD was recently developed. In this class, surfactant can be used alone or in the combination with other hydrophilic carriers in the preparation of SD (Figure 2). Surfactants were widely used to improve the solubility and BA of poorly water-soluble drugs and play a crucial role in the pharmaceutical industry. Adsorption of a surfactant on a solid surface can modify the hydrophobicity of the drug, thereby reducing surface tension between two liquids or between a liquid and a solid. In addition, surfactants can also act as wetting agents, detergents, emulsifiers, foaming agents, and dispersants. Several surfactants such as Inulin [69,70], inutec [70], poloxamer 407 [71], Gelucire 44/14 [72], and Compritol 888 ATO [73] are used in preparation of SD. In a study by Panda et al. [74], Gelucire 50/13 and poloxamer 188 were used in the development of a bosentan SD formulation to improve the solubility and dissolution of this drug. The results showed that the solubility of bosentan from the SD formulation increased 8- and 10-fold when using Gelucire 50/13- and poloxamer 188-based SDs, respectively, in comparison with that of the pure drug. Furthermore, over 90% of the drug was released from SD after 1-h in vitro dissolution studies. Karolewicz et al. [75] prepared an SD of fenofibrate with poloxamer 407 as the carrier at ratios of 10/90, 20/80, 30/70, 40/60, 50/50, 60/40, 70/30, 80/20, and 90/10 using the fusion method. The results showed a 134-fold increase in dissolution rate for SD containing 30/70 *w*/*w* fenofibrate/ poloxamer 407. Surfactants are also used in preparation of SD of poorly soluble anticancer drugs such as docetaxel [76], flutamide [77], and lapatinib [78]. 

### 2.2. Structure-Based Class of Solid Dispersion

#### 2.2.1. Eutectic Mixtures

A eutectic mixture is a mixture of two components that melt at a single temperature. Components A and B were co-melted at the eutectic point (E) (Figure 3), where the melting point of the mixture was lower than that of component A or B alone. In 1961, Sekiguchi and Obi [39] were the first to prepare a eutectic mixture of sulfathiazole and urea. The results showed that the absorption of sulfathiazole in the eutectic mixture was improved compared to that of the conventional drug.

#### 2.2.2. Solid Solution

Herein, SD is a mixture of the drug and a carrier [79]. Solid solution is categorized on the basis of miscibility and molecular size of the components as continuous and discontinuous solid solutions. In continuous solid solutions, the two components can be mixed in all proportions at which the bonding strength between the two components is greater that of the individual components [80]. In discontinuous solid solutions, the solubility of each component is limited in solid solvents [81]. Solid solutions are classified as substitutional (Figure 4A) and interstitial (Figure 4B) based on molecular size. In substitutional solid solutions, solute molecules substitute for solvent molecules in the crystal lattice. In interstitial solid solutions, the dissolved molecules occupy the interstitial spaces between the solvent molecules in the crystal lattice [82].

#### 2.2.3. Glass Solution/Glass Suspension

A glass solution is a homogeneous system in which the drug molecule is dissolved in a glassy solvent [81,82]. Glass suspension is a homogeneous system in which the drug molecule is suspended in a glassy solvent [83]. The glassy state is characterized by transparency and brittleness below the glass transition temperature for both glass solutions and glass suspensions.

### 2.3. Advantages of Solid Dispersions

SD was widely used for enhancing dissolvability in water of poorly water-soluble drugs with several advantages as follows: One of the most important advantages of SD is drugs interacting with hydrophilic carriers can decrease agglomeration and release in a supersaturation state, resulting in rapid absorption and improved BA [84].SD can improve drug wettability and increase the surface area, resulting in enhanced aqueous solubility of drugs.SD can be produced as a solid oral dosage form, which is more convenient for patients than other forms like liquid products.In addition, SD showed an advantage compared to salt formulation, cocrystallization, and other methods. For example, salt formulations use ionized active pharmaceutical ingredients (APIs) (cationic or anionic form) and are widely used in the pharmaceutical industry due to the broad capacity of design according to desired drug properties. However, not all drugs can ionize with all cations/anions, and phase dissociation or stability issue is inherent in salt formation or cocrystallization. Salt formulation showed several disadvantages such as reduced solubility and dissolution rate, resulting in decreased relative BA (common ion effect for HCl salts); greater regulatory scrutiny for strong acid salts isolated from alkyl alcohols; and increased hygroscopicity, e.g., for Na and, K salts, spray-drying/lyophilization can dissociate strong acid salts. The disadvantages of salt formulation can be resolved when the formulation is produced using an SD.Practically, dissolution of drugs is a prerequisite for complete absorption to have the desired therapeutic effect of anticancer drugs after oral administration. Most of the anticancer drugs exhibit poor aqueous solubility causes of dissolution limit resulting low BA and high variability in blood concentration. The limitation of drug dissolution can improve by SD, a technique that induces supersaturated drug dissolution and with that it enhances in vivo absorption.

### 2.4. Disadvantages of Solid Dispersions 

SD is a good technique for improving solubility and BA of hydrophobic drugs. However, some disadvantages are as follows:Physical instability.SDs show changes in crystallinity and decreased dissolution rate with aging.Due to their thermodynamic instability, SD is sensitive to temperature and humidity during storage. These factors can promote phase separation and crystallization of SD by increasing the overall molecular mobility, decreasing the glass transition temperature (*T*_g_) or disrupting interactions between the drug and carrier, resulting in a decreased solubility and dissolution rate of the drug.Patients suffering from cancer should continue to use anticancer drugs during treatment. However, the instability of SD during the period of storage can affect drug quality and the effectiveness of treatment.

### 2.5. Preparation Methods for Solid Dispersions

SD can be prepared by several methods such as solvent evaporation, melting, and supercritical fluid (SCF) technology (Figure 5). The list of drugs investigated for SDs is shown in Table 1, and a list of commercial SDs is shown in Table 2.

#### 2.5.1. Melting Method/Fusion Method

The melting method was first used in 1961 by Sekiguchi and Obi [39]. The basic principle of the melting method is that a physical mixture of a drug and hydrophilic carrier is heated directly until they melt at a temperature slightly above their eutectic point. Then, the melt is cooled and solidified rapidly in an ice bath with stirring. The final solid mass is crushed and sieved. The advantages of this method are simplicity and economy. Several drug SDs have been prepared using this method such as sulfathiazole [39], fenofibrate [75], furosemide [85], albendazole [54], and paclitaxel [86] (Table 1). The melting method has also been used to improve solubility of poorly soluble anticancer drugs. For example, to improve solubility of prednisolone, an SD was prepared by the melting method using PEG 4000 and mannitol as the carriers [87]. The results showed that, at weight ratios of drug: PEG 4000 (1:4) and drug: mannitol (1:7), release of drug from the SD (~85%) increased in comparison with the pure drug (~50%). In a study for improving release of paclitaxel from poly(ε-caprolactone) (PCL)-based-film, an SD of paclitaxel was prepared by the melting method using poloxamer 188 and PEG as the carriers and was then incorporated into PCL films. Drug released from SD was higher than that from the pure drug, with over 90% of drug released from the SD after 1 h at a weight ratio of drug: poloxamer 188 (1:3).

#### 2.5.2. Solvent Evaporation Method

The solvent evaporation method is one of the most commonly used methods in the pharmaceutical industry for improving solubility of poorly water-soluble drugs. This method was developed mainly for heat unstable components because drug and carrier are mixed by a solvent instead of heat as in melting method. Therefore, this method allows use of carriers with an excessively high melting point. The basic principle of this method is that drug and carrier are dissolved in a volatile solvent for homogeneous mixing. SD is obtained by evaporating the solvent under constant agitation. Then, the solid SD is crushed and sieved. This method was first applied by Tachibana and Nakamura in 1965 [131]. The formulation was prepared by dissolving a drug (β-carotene) and a carrier (PVP) in an organic solvent (chloroform). After that, the solvent was completely evaporated to form a solid mass, which was then sieved and dried. The main advantage of this method is avoidance of decomposition of drug and carrier because the required temperature for evaporation is low. In 1966, Mayersohn and Gibaldi developed an SD of griseofulvin using PVP as the carrier and chloroform as the solvent [132]. Dissolution of griseofulvin from the SD was 11-times greater than that of the pure drug at a ratio of griseofulvin: PVP (1:20). This method has been used to improve solubility of many drugs such as azithromycin [91], tectorigenin [92], flurbiprofen [93], cilostazol [94], ticagrelor [95], piroxicam [96], indomethacin [97], loratadine [98], diclofenac [68], abietic acid [99], efavirenz [100], and repaglinide [101] (Table 1). An SD of tectorigenin, PVP, and PEG 4000 at a weight ratio of 7:54:9 was prepared using solvent evaporation to increase dissolution and BA [92]. In vitro release of the drug from the SD was 4.35-fold greater than that of the pure drug after 2.5 h. In addition, the oral BA of the drug from the SD was higher than that of the conventional drug as determined by AUC (4.8-fold) and C_max_ (13.1-fold). The solvent evaporation method has often been used to improve solubility of poorly water-soluble anticancer drugs such as paclitaxel [133], docetaxel [76], and others (Table 3). For example, the solubility and dissolution of emulsified SD of docetaxel at 2 h were 34.2- and 12.7-fold greater, respectively, compared to those of the conventional drug [76]. In the study by Adeli [91], azithromycin SD was prepared by the solvent evaporation method using various PEG such as PEG 4000, PEG 6000, PEG 8000, PEG 12,000, and PEG 20,000 as the carriers at different ratios. Using PEG as the hydrophilic carrier in SD, the solubility of drug is improved compared to the pure conventional drug. The best result was obtained from SD containing azithromycin: PEG 6000 (1:7). After 1 h, the amount of azithromycin released from SD was more than 49%. 

#### 2.5.3. Melting Solvent Method (Melt Evaporation)

The melting solvent method was first studied by Goldberg el al. [145]. In their study, an SD was prepared to improve dissolution of griseofulvin using succinic acid as the carrier and methanol as the solvent. The melting solvent method combines melting method and solvent evaporation method. The drug is first dissolved in a suitable solvent and incorporated into the melt of the carrier, and the mixture is then evaporated to dryness. Practically, this method is very useful for drugs with a high melting point. Chen el al. [146] showed a novel monolithic osmotic tablet composed of an SD of 10-hydroxycamptothecin (HCPT) prepared by the melting solvent method with PEG 6000 as the carrier and methanol as the solvent. At 12 h, the cumulative release of drug was over 90%, and the optimized formulation was able to deliver HCPT at a constant rate of 1.21 mg/h for 12 h in simulated intestinal fluid (SIF; pH 6.8).

#### 2.5.4. Melt Agglomeration Process

Melt agglomeration is a process in which a binder acts as a carrier. In this method, the drug, binder, and other excipients are heated to above the melting point of the binder. Alternatively, a dispersion of the drug is sprayed onto the heated binder [147,148,149,150]. A diazepam SD was prepared by melt agglomeration method in a high shear mixer to improve the dissolution rate. In this preparation, lactose monohydrate was used as the binder and was melt agglomerated with PEG 3000 or Gelucire 50/13. The binder was added by either pump-on or melt-in procedures. Use of melt agglomeration resulted in a high dissolution rate at a lower drug concentration. The dissolution rates were similar between pump-on and melt-in procedures. In addition, the SD of diazepam containing Gelucire 50/13 showed higher dissolution compared with the SD containing PEG 3000.

#### 2.5.5. Hot-Melt Extrusion Method

Hot-melt extrusion is a common method for improving solubility and oral BA of poorly water-soluble drugs, in which the amorphous SD is formed without solvent, thereby avoiding residual solvents in the formulation [151]. This method is conducted by a combination of the melting method and an extruder, in which a homogeneous mixture of drug, polymer, and plasticizer is melted and then extruded through the equipment. The shapes of products at the outlet of extruder can be controlled and do not require grinding in the final step. For example, the melt extrusion method was used to increase dissolution and oral BA of oleanolic acid [103]. Using PVP VA 64 as the carrier, an SD of oleanolic acid was successfully prepared. Dissolution of this SD was better (about 90% of drug from SD released in the 10 min) in comparison with those of a physical mixture (45% after 2 h) and pure drug (37% after 2 h). In addition, the AUC_0–24h_ (1840 ± 381.8 ng·h/mL) and C_max_ (498.7 ng/mL) of the drug from the SD were enhanced 2.4 times and 5.6 times compared with those of pure drug (761.8 ± 272.2 ng·h/mL and 89.1 ± 33.1 ng/mL, respectively). In another study by Sathigari el al., [104], efavirenz SD was prepared via hot-melt extrusion method using Eudragit EPO or Plasdone S-630 as carriers to improve the dissolution rate of efavirenz. In the dissolution test, because of very low aqueous solubility (3–9 µg/mL), sodium lauryl sulfate (SLS) was added in the dissolution medium. The results showed that the solubility of efavirenz increased substantially (197 µg/mL) in comparison with that of the pure drug. After 30 min, about 96% and 82% of drug was released from SD using Eudragit EPO and Plasdone S-630 as carriers, approximately 2-fold and 1.7-fold higher compared with drug alone, respectively. In addition, the SD was stable after 9 months.

#### 2.5.6. Lyophilization Techniques/Freeze-Drying

Lyophilization is an alternative process to the solvent evaporation method in which the drug and carrier are dissolved in a solvent and then the solution is frozen in liquid nitrogen to form a lyophilized molecular dispersion [152]. This method is typically used for thermolabile products that are unstable in aqueous solutions but stable in the dry state for prolonged storage periods. In a previous study, nifedipine and sulfamethoxazole SD were prepared by Soluplus and PEG 6000 as carriers to evaluate physicochemical and in vitro characteristics [112]. SDs of the two drugs were successfully prepared, and drug dissolution rate were increased. In a study of the anticancer drug exemestane [138], an exemestane-loaded phospholipid/sodium deoxycholate SD was prepared to improve the solubility and oral BA of the drug. The solubility and dissolution rate of exemestane from SD were increased compared to those of the pure drug. The absorptive transport of the SD was 4.6-fold greater in comparison with that of the conventional drug. Furthermore, the AUC_0–72h_ of the drug in the SD was 2.3-fold greater than that of the drug alone. In another study on a flutamide SD prepared by lyophilization to enhance the dissolution rate [77], PVP K30, PEG 6000, and poloxamer 407 were used as the carriers. Among these carriers, dissolution of SD when using poloxamer 407 as the carrier (81.6%) was higher compared with that using other carriers (PVP K30 66.5% and PEG 6000 78.2%) after 30 min and higher compared with that of the pure drug (13.5%). 

#### 2.5.7. Electrospinning Method

The electrospinning method is a combination of SD technology and nanotechnology. In this method, solid fibers are produced from a polymeric fluid stream or melt delivered through a millimeter-scale nozzle [153]. The advantage of this method is that the process is simple and inexpensive. This method is suitable for preparing nanofibers and controlling release of biomedicine. A nanofiber of polyvinyl alcohol (PVA):ketoprofen (1:1, *w*/*w*) was prepared by the electrospinning method [154]. Dissolution of this nanofiber was significantly (*p* < 0.05) greater than that of ketoprofen alone. In another study, an amorphous formulation of indomethacin and griseofulvin was prepared by the electrospinning method using PVP as the carrier. This formulation was stable for 8 months in a desiccator [155].

#### 2.5.8. Co-Precipitation 

In this method, the carrier is first dissolved in solvent to prepare a solution, and the drug is incorporated into the solution with stirring to form a homogeneous mixture. Then, water is added dropwise to the homogenous mixture to induce precipitation. Finally, the precipitate is filtered and dried. In a study by Sonali et al. [116], a silymarin SD was prepared with HPMC E15LV as the carrier with various methods such as kneading, spray-drying, and co-precipitation. The silymarin SD prepared by co-precipitation showed significantly (*p* < 0.05) enhanced dissolution compared with the other two methods. Furthermore, the solubility of silymarin from the SD prepared by co-precipitation improved 2.5-fold in comparison with that of the conventional drug. 

#### 2.5.9. Supercritical Fluid (SCF) Technology

SCF was introduced in late 1980s and early 1990s. SCF produces a formulation with a narrow particle size range (microparticles or nanoparticles) without solvent and was reported by Hannay and Hogarth in 1897 as a medium for particle production [156]. A substance is in the supercritical state when the temperature and pressure are above its critical point. SCF can act as solvent or antisolvent in SD. The basic principle of SCF is that the drug and carrier are dissolved in a supercritical solvent (e.g., CO_2_) and sprayed through a nozzle into an expansion vessel with lower pressure. The rapid expansion induces rapid nucleation of the dissolved drugs and carriers, leading to the formation of SD particles with a desirable size distribution in a very short time. To date, SCF can be performed by several methods such as rapid expansion from supercritical solution (RESS) [157], gas antisolvent (GAS) [141], supercritical antisolvent (SAS) [119], and solution enhanced dispersion by SCF (SEDS) [158]. The RESS process is conducted as follows: Drug and carrier are dissolved in SCF, then sprayed through an atomizer in an expansion vessel maintained at low pressure, resulting in formation of an SD. The advantage of this method is that it can minimize use of organic solvents for preparation of SD. In SCF technology, CO_2_ is a suitable solvent for preparation of SD of insoluble drugs, primarily due to its low critical temperature (31.04 °C) and low critical pressure (7.38 MPa), lack of toxicity, lack of inflammability, and environmental safety [159]. To improve dissolution of irbesartan, Adeli [119] prepared an SD by the SAS method using poloxamer 407 as the carrier. The optimal ratio of drug and carrier was 1:1. As a result, dissolution of the irbesartan-SAS sample was 13 times higher than that of the pure drug. In another study, to enhance the BA of apigenin, apigenin nanocrystals were prepared by the SAS method [120]. The results showed that the C_max_ and AUC of the final formulation increased 3.6-fold and 3.4-fold, respectively, in comparison with that of the drug alone, demonstrating improved BA. In drug development, SCF technology is a potential method for enhancing solubility and BA of poorly water-soluble drugs. One limitation of this method is that most drugs are not soluble in CO_2_. 

#### 2.5.10. Spray-Drying Method

Spray-drying is one of the oldest methods for drying materials, especially thermally-sensitive materials such as foods and pharmaceuticals. In this method, the drug is dissolved in a suitable solvent, and the carrier is dissolved in water to prepare the feed solution. Then, the two solutions are mixed by sonication or other suitable methods until the solution is clear. In the procedure, the feed solutions were firstly sprayed in a drying chamber via a high-pressure nozzle to form fine droplets. The formed droplets are composed of drying fluid (hot gas) and form particles of nano or micro size [160]. Clinically, the spray-drying method has been widely used for preparation of SD for improving solubility and BA of poorly water-soluble drugs such as nilotinib [124], spironolactone [125], valsartan [126], rebamipide [127], and artemether [128] (Table 1). For example, in a study by Herbrink et al. [124], an SD of nilotinib was prepared by spray-drying to enhance solubility. Soluplus was selected as the best carrier based on in vitro dissolution studies. At a drug: Soluplus (1:7) ratio, the solubility of nilotinib was improved 630-fold in comparison with the pure drug. In another study by Pawar et al. [128], an artemether SD was prepared by spray-drying to improve solubility and dissolution. The results showed that the optimal ratio of drug: carrier (artemether: Soluplus) was 1:3. After 1 h, artemether release from SD was 82%, 4.1-fold higher than the conventional drug (20%). Spray-drying is an efficient technology for preparation of SD for improving solubility and BA of hydrophobic drugs.

#### 2.5.11. Kneading Method

In this method, the carrier is dispersed in water and processed into a paste. Then, the drug is added and kneaded thoroughly. The final kneaded formulation is dried and passed through a sieve if necessary. In a previous study by Dhandapani and El-gied [67], cefixime SD was prepared with β-CD as the carrier using the kneading method. The result showed that the dissolution rate of cefixime from the SD was 6.77-fold greater than that of the pure drug, suggesting a possible improvement in BA. In another study [130], the HP-β-CD was used as the carrier in a domperidone SD. Saturation solubility and in vitro dissolution of domperidone from SD were considerably higher (3-fold) in comparison with the pure drug.

#### 2.5.12. Suitable Methods for Production of SDs of Anticancer Drugs

Anticancer drugs are classified into biologic drugs (monoclonal antibodies) and small molecule drugs (nonbiologic anticancer drugs) based on effectiveness and safety profile. Biologic drugs are administered by intravenous (IV) injection due to their large molecular weight, while small molecule drugs are preferentially administered by gastrointestinal route. Oral administration is currently preferred for treatment of cancer in comparison with IV route because it is convenient, painless, safe, and economic. Oral drugs can be administered at home, do not induce the same discomfort as an IV infusion, and the drug concentration can be maintained for long time periods in cancerous cells. In addition, oral dosage forms are easy to store and transport. Therefore, oral administration has received increasing attention, leading to increased numbers of anticancer drugs being developed for oral dosing. 

To date, SD technology is widely used to improve the solubility and BA of anticancer drugs due to its simplicity, economy, and high effectiveness. Most methods are suitable for making SD, in which melting method, solvent evaporation method, SCF technology, and freeze-drying are common for production of SD formulation of anticancer drugs in comparison with other methods. The selected method will be based on physicochemical properties of anticancer drugs.

#### 2.5.13. Lab Scale and Industrial Scale Manufacturing Processes

Several manufacturing methods were used to produce SD. However, not all methods are available for commercial processes. Practically, the melting method and solvent evaporation method are two distinct processes that are widely used on lab and industrial scale. 

On the lab scale, for the solvent evaporation method, a rotary evaporator was mostly used to produce SD. Recently, SCF and freeze-drying are also employed. Due to its simplicity and economy, the melting method is popularly used. Currently, several types of equipment from many manufacturers such as Brabender Technologies, Coperion GmbH, Thermo Fisher Scientific, and Leistritz Advanced Technologies Corp are available in the laboratory, in which SD amount can be produced from a few grams to a kilogram. 

On the industrial scale, production of SD is not as simple as at the lab scale because it involves a large amount of product from a few to several hundred kilograms. In addition, processes need to be robust, reproducible, and follow good manufacturing practices (GMP). These are difficult to ensure for processes such as solvent cast evaporation or water bath melting process. Spray-drying and freeze-drying are the most representative of the solvent evaporation methods used for manufacturing SD. Moreover, the spray-drying process is easy to scale up from lab scale to industrial scale. Melt agglomeration and hot-melt extrusion are two types of melting processes available on the industrial scale. For instance, hot-melt extrusion is one of the most common methods used on an industrial scale to produce SD using twin-screw extruder with a large diameter of the screw (16–50 mm) compared with small diameter of the screw at lab scale (11–16 mm). 

In summary, the selected method for manufacturing process plays an important role in the success of a formulation. On the lab scale, the criteria for selecting the melting method are based on the melting point and thermal stability. For selecting the solvent evaporation method, important factors to consider are properties of the drug, carrier, and an organic solvent. On the industrial scale, the production of SD is limited to only a few manufacturing processes. Hot-melt extrusion is the most common among the melting processes to produce SD. For the evaporation method, the selection criteria are based on solvent toxicity and loading capacity.

### 2.6. Use of SD for Improving Poorly Soluble Anticancer Drugs

Cancer is one of the leading causes of death worldwide and is defined as a group of diseases involving abnormal cell growth with potential to invade or spread to other parts of the body. The World Health Organization predicted that the burden of cancer will increase to 23.6 million new cases each year by 2030 [161]. In 2018, among 1,735,350 new cancer cases, an estimated 609,640 Americans will die from cancer, corresponding to almost 1,700 deaths per day [162]. Therefore, treatment of cancer is one of the most important issues studied during the past several decades. For anticancer drugs to induce a therapeutic effect, they must first be absorbed and enter the circulation. To ensure complete BA, most anticancer drugs are preferably administered by IV infusion because the entire dose of the drug will directly enter into the circulatory system and instantaneously distribute to its sites of action. However, IV administration inconveniences the patients because they have to visit the hospital to receive treatment. In addition, several side effects may occur during the treatment period. For example, the commercial product paclitaxel (Taxol), which is prepared with Cremophor EL and ethanol as solvents (50:50, *v*/*v*), is associated with serious side effects due to Cremophor EL such as severe hypersensitivity, myelosuppression, neutropenia, and neurotoxicity [16,17,18]. To avoid these side effects, Park et al. [163] prepared paclitaxel SD without Cremophor EL using the supercritical antisolvent method. The solubility of paclitaxel in SD is 10 mg/mL, an almost 10 000-fold increase compared to the conventional drug at a weight ratio of 1/20/40 of paclitaxel/HP-β-CD/HCO-40. In addition, the SD is stable over 6 months. 

In the past few decades, many oral formulations of anticancer drugs have been developed. Oral administration is currently the desired route for treatment of cancer because it is convenient, painless, safe, and economic. In addition, oral dosage forms are easy to store and transport. The prerequisite for oral administration is complete and predictable absorption. To achieve this, drugs have to dissolve in water to absorb in GI tract to be effectively taken up in the circulatory system. However, nearly all anticancer drugs are poorly water-soluble, which can lead to incomplete absorption and poor BA, resulting in large inter- and intra-individual variability in drug concentrations in vivo. Thus, improving the solubility of anticancer drugs is a great challenge in development of improved cancer therapies in the pharmaceutical industry [164]. Among several methods such as complexation, lipid-based systems, micronization, nanonization, and co-crystals, SD is the most successful for improving solubility and BA of anticancer drugs. Vemurafenib (Zelboraf^®^ Roche), regorafenib (Stivarga^®^, Bayer), and everolimus (Afinitor^®^, Votubia^®^, Certican^®^, Novartis) are three commercial anticancer drugs that were prepared by SD [165]. Zelboraf^®^ was prepared from vemurafenib and hypromellose acetate succinate carrier at a weight ratio of 30/70 (*w*/*w*). Dissolution of the SD formulation was approximately 30 times higher compared to the crude powder. Stivarga, which contains regorafenib and PVP-25 as the carrier, showed a 4.5-fold increase in dissolution rate compared to that of the drug mixture. In addition, the BA of the drug from the SD was approximately 7 times higher than that of the conventional drug. Afinitor is an SD prepared with drug and HPMC at a weight ratio of 1:40 in which the dissolution rate from the SD was improved about 4-fold in comparison with the pure drug. The anticancer drugs investigated for SD are shown in Table 3. To enhance the solubility and dissolution rate of docetaxel, an SD was prepared and the solubility and dissolution rate at 2 h were 34.2- and 12.7-fold higher in comparison with the crude powder, respectively [76]. In another study by Ren et al. [134], dissolution of bicalutamide was improved by preparing an SD using PVP K30 as the carrier at a weight ratio of drug: PVP K30 (1:5). At this ratio, about 98% of bicalutamide was dissolved during the first 10 min. Recently, mixtures of surfactants were used to prepare SD, resulting in increased solubility and improved permeability of BCS Class IV drugs. Song et al. [135] prepared a docetaxel SD using poloxamer F68 alone or a poloxamer SD using a combination of poloxamer F68 and poloxamer P85. Performance of the two SDs was compared, showing that the SD prepared with only poloxamer F68 increase in solubility (1.39-fold increases in BA), while the SD prepared with a combination of poloxamer F68 and poloxamer P85 showed enhanced solubility and permeability (2.97-fold increase in BA). Thus, SD is a promising technique for improving solubility and BA of poorly water-soluble anticancer drugs.

### 2.7. Future Prospects

SD is currently considered one of the most effective methods for enhancing the solubility and BA of poorly water-soluble drugs. Even though the issues related to preparation, stability, and storage formulation of drugs may limit the numbers of commercial SD products on the market, SD products are still steadily increasing in clinical settings based on improved manufacturing methods and carriers to solve the above problems.

In recent years, carriers used in the preparation of SD have been developed. Some studies used new carriers, while other studies used more than one carrier for production of SD formulation. Using more than one carrier in the formulation of SD, many effective methods were designed, recrystallization was decreased, and the stability of SD was improved. Some carriers used recently are Inulin^®^, Gelucire^®^, Pluronic^®^, and Soluplus^®^. 

In the manufacturing process, Kinetisol Dispersing (KSD) [166,167,168] is a novel high-energy mixing process for preparation of SD, in which the drug and carrier are processed by utilizing a series of rapidly rotating blades through a combination of kinetic and thermal energy without the aid of external heating sources. This brings new hope for development of more SD products in the future. 

## 3. Conclusions

In this review, we focused on classification of SD, methods for preparation of SD, and current trends in SD for improving the solubility of poorly water-soluble drugs, including anticancer drugs. IV administration is preferred for anticancer treatment. However, patients are inconvenienced by this route because they have to visit a hospital to receive treatment. Therefore, scientists are working to develop oral dosage forms of anticancer drugs. Oral administration is currently the most common route of administration of drugs because it is convenient for patients. A prerequisite for oral administration is dissolution of the drug in water to allow absorption in the GI tract; however, approximately 40% of NCEs including anticancer drugs are insoluble in water which leads to poor absorption, poor BA, and high intra- and inter-individual variability in blood concentrations. Therefore, improving the solubility of poorly water-soluble drugs is a large challenge in the pharmaceutical industry. To overcome this problem, various methods such as complexation, lipid-based systems, SD, micronization, nanonization, and cocrystallization were developed for clinical use. Among these, SD is one of the most successful methods and is widely used for development of drugs. It is considered a promising technique to overcome problems related to poor aqueous solubility and poor BA. By improving wettability of drugs and surface area, drug solubility and dissolution were increased. In the preparation of SD, understanding the properties of the carrier and drug, and selecting a suitable method play crucial roles in the success of the formulation. 

## Figures and Tables

**Figure 1 pharmaceutics-11-00132-f001:**
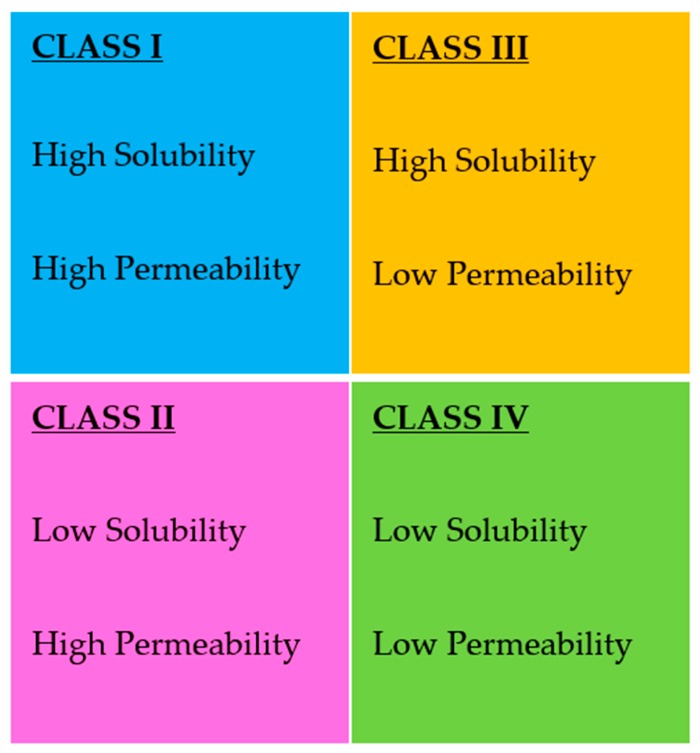
Biopharmaceutics classification system (BCS).

**Figure 2 pharmaceutics-11-00132-f002:**
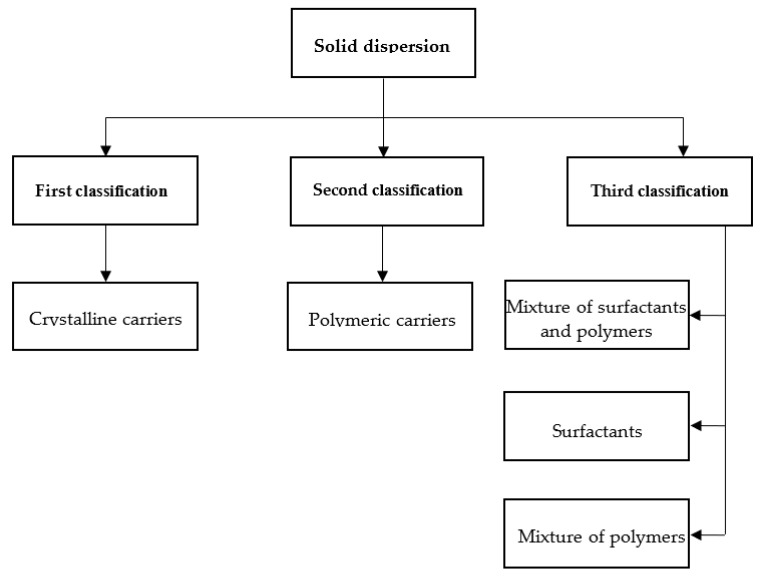
Classification of solid dispersions.

**Figure 3 pharmaceutics-11-00132-f003:**
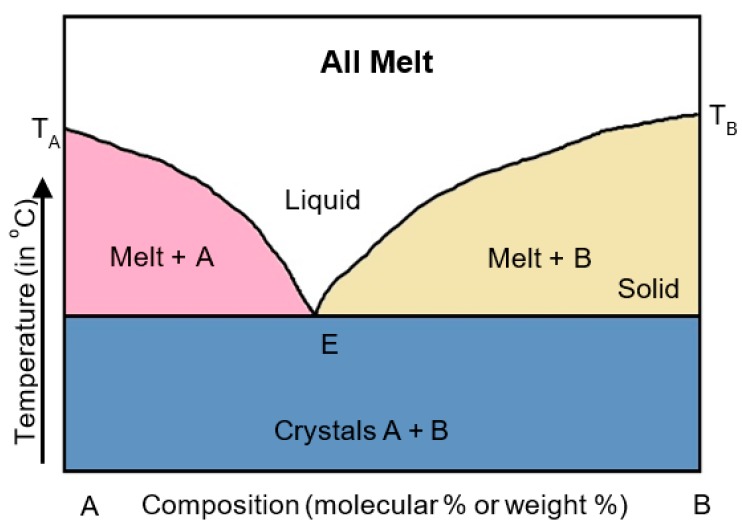
Phase diagram of a eutectic mixture. A, B (drug, carrier), E (eutectic point).

**Figure 4 pharmaceutics-11-00132-f004:**
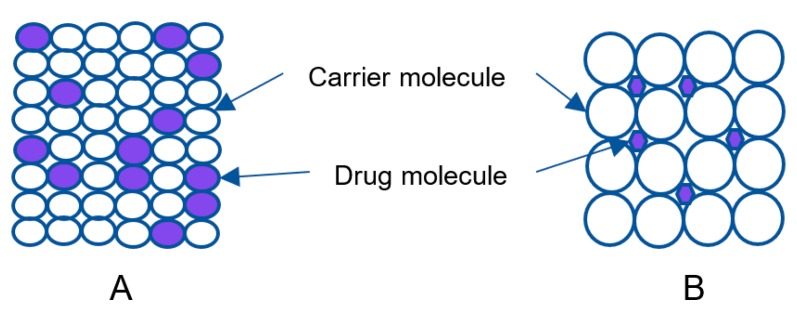
Schematic structure of the solid solution.

**Figure 5 pharmaceutics-11-00132-f005:**
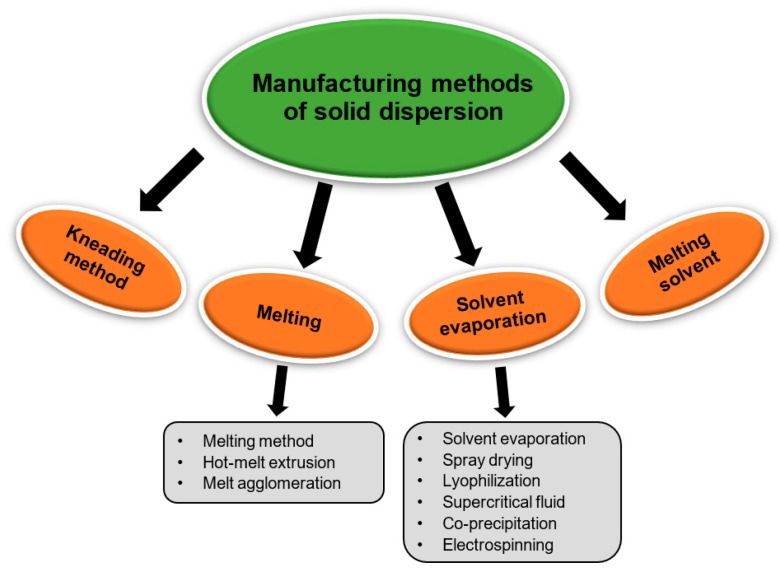
Manufacturing methods of solid dispersion.

**Table 1 pharmaceutics-11-00132-t001:** List of drugs investigated for solid dispersions.

Methods	Drugs
Melting/fusion method	Sulfathiazole [39], clotrimazole [43], albendazole [54], tacrolimus [61], fenofibrate [75], furosemide [85], paclitaxel [86], manidipine [88], olanzapine [89], diacerein [90]
Solvent evaporation method	Dutasteride [23], tadalafil [50], glimepiride [53], nimodipine [59], diclofenac [68], azithromycin [91], tectorigenin [92], flurbiprofen [93], cilostazol [94], ticagrelor [95], piroxicam [96], indomethacin [97], loratadine [98], abietic acid [99], efavirenz [100], repagnilide [101], prednisolone [102]
Hot-melt extrusion method	Ritonavir [37], naproxen [46], oleanolic acid [103], efavirenz [104], tamoxifen [105], lafutidine [106], disulfiram [107], bicalutamide [108], itraconazole [109], miconazole [110], glyburide [111]
Lyophilization/Freeze-drying	Nifedipine and sulfamethoxazole [112], celecoxib [113], meloxicam [114], docetaxel [115]
Co-precipitation method	Silymarin [116], celecoxib [117], GDC-0810 [118]
Supercritical fluid method	Ketoprofen [66], irbesartan [119], apigenin [120], carbamazepine [121], glibenclamide [122], carvedilol [123]
Spray-drying method	Nilotinib [124], spironolactone [125], valsartan [126], rebamipide [127], artemether [128], naproxen [129]
Kneading method	Cefixime [67], efavirenz [100], domperidone [130]

**Table 2 pharmaceutics-11-00132-t002:** List of commercial solid dispersions.

Products	Drugs	Polymers	Company
Afeditab^®^	Nifedipine	Poloxamer or PVP	Elan Corp, Ireland
Cesamet^®^	Nabilone	PVP	Lilly, USA
Cesamet^®^	Nabilone	PVP	Valeant Pharmaceuticals, Canada
Certican^®^	Everolimus	HPMC	Novartis, Switzweland
Gris-PEG^®^	Griseofulvin	PEG	Novartis, Switzweland
Gris-PEG^®^	Griseofulvin	PVP	VIP Pharma, Denmark
Fenoglide^®^	Fenofibrate	PEG	LifeCycle Pharma, Denmark
Nivadil^®^	Nivaldipine	HPC/HPMC	Fujisawa Pharmaceuticals Co., Ltd
Nimotop^®^	Nimodipine	PEG	Bayer
Torcetrapib^®^	Torcetrapib	HPMC AS	Pfizer, USA
Ibuprofen^®^	Ibuprofen	Various	Soliqs, Germany
Incivek^®^	Telaprevir	HPMC AS	Vertex
Sporanox^®^	Itraconazole	HPMC	Janssen Pharmaceutica, Belgium
Onmel^®^	Itraconazole	HPMC	Stiefel
Prograf^®^	Tacrolimus	HPMC	Fujisawa Pharmaceuticals Co., Ltd
Cymbalta^®^	Duloxetine	HPMC AS	Lilly, USA
Noxafil^®^	Posaconazole	HPMC AS	Merck
LCP-Tacro^®^	Tacrolimus	HPMC	LifeCycle Pharma, Denmark
Intelence^®^	Etravirine	HPMC	Tibotec, Yardley, PA
Incivo^®^	Etravirine	HPMC	Janssen Pharmaceutica, Belgium
Rezulin^®^	Troglitazone	PVP	Pfizer, USA
Isoptin SRE-240^®^	Verapamil	Various	Soliqs, Germany
Isoptin SR-E^®^	Verapamil	HPC/HPMC	Abbott Laboratories, USA
Crestor^®^	Rosuvastatin	HPMC	AstraZeneca
Zelboraf^®^	Vemurafenib	HPMC AS	Roche
Zortress^®^	Everolimus	HPMC	Novartis, Switzweland
Kalydeco^®^	Ivacaflor	HPMC AS	Vertex
Kaletra^®^	Lopinavir and Ritonavir	PVP/polyvinyl acetate	Abbott Laboratories, USA

PVP: polyvinylpyrrolidone; HPMC: hydroxypropylmethylcellulose; PEG: polyethyleneglycol; HPC: hydroxypropylcellulose; HMPC AS: hydroxypropylmethylcellulose acetylsuccinate.

**Table 3 pharmaceutics-11-00132-t003:** Anticancer drugs investigated for solid dispersions.

Anticancer Drugs	Carriers	Methods	Attributes of Modified Anticancer Drugs	Reference	Years
Bicalutamide	PVP K30	Solvent evaporation	Using PVP K30 as carrier, SD showed the highest cumulative released percentage (about 98% during the initial 10 min) and stability after 6 months	[134]	2006
Docetaxel	HPMC, PEG	Solvent evaporation	The solubility and dissolution of emulsified SD of docetaxel at 2 h were 34.2- and 12.7-fold higher, respectively, compared to the pure conventional drug	[76]	2011
Docetaxel	Poloxamer F68/P85	Freeze-drying	A combination of poloxamer F68 and P85 in the preparation of docetaxel SD not only enhanced solubility, but also improved intestinal permeation	[135]	2016
Etoposide	PEG	Fusion method	The solubility and dissolution of etoposide in SD were higher in comparison with etoposide alone	[136]	1993
Everolimus	HPMC	Co-precipitation	At a ratio of drug to HPMC (1:15), drug release from SD was 75% after 30 min, thereby improving oral absorption of everolimus	[137]	2014
Exemestane	Lipoid^®^ E80S/sodium deoxycholate	Freeze-drying	The exemestane SD showed 4-6-fold increase in absorptive transport compared to the pure drug. In addition, AUC_0-72h_ of exemestane SD was 2.3-fold higher in comparison with that of drug alone	[138]	2017
Flutamide	PVP K30, PEG, Pluronic F127	Lyophilization	The dissolution of flutamide was higher (81.64%) than the drug alone (13.45%) using poloxamer 407 as a carrier	[77]	2010
Lapatinib	Soluplus, poloxamer 188	Solvent evaporation, hot-melt extrusion	Solubility and dissolution of lapatinib SD were enhanced compared to the drug alone. After 15 min, the drug in SD was released at 92%compared to the drug alone (48%)	[78]	2018
Letrozole	CO_2_-menthol	Supercritical fluid	Solubility of letrozole SD using supercritical fluid is 7.1 times higher compared to that of the conventional drug	[139]	2018
Megestrol acetate	HPMC, Ryoto sugar ester L1695	Supercritical fluid	The SD with drug: HPMC: Ryoto sugar ester L1695 ratio of 1:2:1 showed over 95% rapid dissolution within 30 min. In addition, AUC and C_max_ (0-24h) of drug in SD were 4.0- and 5.5-fold higher, respectively, compared to those in pure drug	[140]	2015
Oridonin	PVP K17	Supercritical fluid	The dissolution of oridonin SD significantly increased compared to the original drug. In addition, the absorption of oridonin in SD showed 26.4-fold improvement in BA	[141]	2011
Paclitaxel	Poloxamer 188, PEG	Fusion method	Paclitaxel SD was successfully prepared, and the drug release from SD was higher than that of the drug alone	[86]	2013
Paclitaxel	HPMC AS	Solvent method	The solubility and permeability of paclitaxel were not increased simultaneously through supersaturation in vivo	[133]	2018
Prednisolone	HP-β-CD, PEG, PVP, PEG 4000, MNT, SMP, Cremophor	Solvent evaporation, melting method, kneading method	The in vitro dissolution of prednisolone SD was improved compared with the pure drug	[87]	2011
Raloxifene	PVP K30	Spray-drying	The absorption of raloxifene from SD showed 2.6-fold enhanced BA in comparison with the conventional drug	[142]	2013
Sorafenib	Soluplus	Spray-drying	The C_max_ and AUC_0-48h_ of sorafenib in SD formulation increased 1.5- and 1.8-fold, resocetuvely, compared with the pure drug	[143]	2015
Tamoxifen	Soluplus	Hot-melt extrusion	The dissolution and BA of tamoxifen in SD were improved compared with the drug alone	[105]	2018
Vemurafenib	HPMC AS	Solvent-controlled precipitation	The BA of vemurafenib in SD was improved 4~5-fold compared to the conventional drug	[144]	2013

HP-β-CD: hydroxypropyl-β-cyclodextrin, MNT: mannitol, SMP: skimmed milk powder.

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
