# Peer review of "Overview of the Manufacturing Methods of Solid Dispersion Technology for Improving the Solubility of Poorly Water-Soluble Drugs and Application to Anticancer Drugs"

_pharmaceutics, 2019, doi:10.3390/pharmaceutics11030132_

Round 1
Reviewer 1 Report
In this manuscript, the authors attempted to review methods aimed at improving solubility of poorly water soluble active molecules. However, first and foremost, this manuscript lacks focus. From reading it, it is very apparent that the authors wanted to concentrate on “solid dispersions of poorly soluble anticancer drugs”, but the title and part of the content do not reflect this. There are many reviews on this very general topic and a review directed towards anticancer drugs will be very beneficial as research efforts mostly concentrate on model, low molecular weight, cheap molecules with no safety issues. The authors also need to differentiate between new information presented in their paper and that shown in another review in this area “Inventory of oral anticancer agents: Pharmaceutical formulation aspects with focus on the solid dispersion technique” (Ref. 141). As it stand, the manuscript in the current form is rather confusing and presents limited new contribution to the field.
So, with the above general and below specific comments in mind, the authors should re-write the paper, change the title and highlight much better the aspects relevant for anticancer drugs.
1. Abstract should clearly present the focus of the paper; outline the key issues with anticancer molecules including a clear division between small molecules (APIs) and biologics, methods that have been discussed in the body of the paper and conclusions/future directions.
2. Section 1. should be named “Introduction” and not “Introductions”.
3. Overall, the authors should improve the English throughout. Please give the manuscript to a native speaker – this review does not aim at correcting the language mistakes.
4. What is missing in your Introduction is a review of the anticancer molecules from the point of view of their physicochemical characteristics highlighting the key property: poor solubility. This is one of the weak aspects of Introduction, on the other hand there is a large section on nanoparticles with only a tangential relevance to the topic.
5. Continuing with Introduction – describing the BCS system without briefly saying what makes the molecule to fall into the 4 classes: permeability (85% is the cut-off) and solubility (pH dependent, highest dose, 250 ml water etc.) is meaningless. You need to expand on this.
Section 2 “Solid dispersion”
6. Solid dispersions are not a manufacturing method (going by the current title), it is an approach/technology and we can utilise e.g. quench cooling, spray drying, freeze drying as manufacturing methods to produce solid dispersions. So please think about an appropriate title for this manuscript.
7. I am sorry, but do not understand your “classification” described in 2.1 and also Figure 2 is not clear at all. Also, how this classification relates to anticancer drugs?
8. Again, Section 2.2 lacks examples of anticancer drugs.
9. Sections 2.3 and 2.4 should include advantages and disadvantages specific to anticancer drugs. At the moment these two sections are not informative and bring nothing new to this paper.
10. Table I is redundant unless is contains anticancer molecules, but those are already presented in Table II.
11. For Section 2.5 there need to be a summary section as which of the method(s) is/are better for making solid dispersions of anticancer drugs. It needs to include aspects of health and safety, exposure to those molecules during production, routes of delivery, formulation etc. Also – it is of a key importance to highlight the differences in processing APIs versus biologics.
Author Response
1. Abstract should clearly present the focus of the paper; outline the key issues with anticancer molecules including a clear division between small molecules (APIs) and biologics, methods that have been discussed in the body of the paper and conclusions/future directions.
The reviewer’s point is very well appreciated.
In response to reviewer’s comment, we added more sentences in the abstract as follows:
Line 15~19: Anticancer drugs are classified into biologic drugs (monoclonal antibodies) and small molecule drugs (nonbiologic anticancer drugs) with a difference in their effectiveness and safety profiles. Biologic drugs are administered by intravenous (IV) injection due to their large molecular weight while small molecule drugs are preferentially by gastrointestinal route.
2. Section 1. should be named “Introduction” and not “Introductions”.
In response to reviewer’s comment, “Introductions” was changed to “Introduction”.
3. Overall, the authors should improve the English throughout. Please give the manuscript to a native speaker – this review does not aim at correcting the language mistakes.
Manuscript will be rechecked as reviewer’s comment.
4. What is missing in your Introduction is a review of the anticancer molecules from the point of view of their physicochemical characteristics highlighting the key property: poor solubility. This is one of the weak aspects of Introduction, on the other hand there is a large section on nanoparticles with only a tangential relevance to the topic.
In response to reviewer’s comment, we added the sentence in the “Introduction” as follows:
Line 86~88: Despite the continuing interest on anticancer drugs in recent years, their use in clinical anticancer therapy is limited due to nonspecific biodistribution, low therapeutic indices and poor aqueous solubility.
5. Continuing with Introduction – describing the BCS system without briefly saying what makes the molecule to fall into the 4 classes: permeability (85% is the cut-off) and solubility (pH dependent, highest dose, 250 ml water etc.) is meaningless. You need to expand on this.
The reviewer’s point is very well appreciated.
Following the reviewer’s comment, we added more information about BCS system in “Introduction” as follows:
Line 100~107: A drug substance is considered highly soluble when the highest single therapeutic dose is soluble in 250 mL or less of aqueous media over the pH range of 1.2 – 6.8 at 37 ± 1°C. Permeability is evaluated on the basis of the extent of absorption of a drug from human pharmacokinetic studies. Alternatively, in vitro culture methods can also use to predict drug absorption in humans. A drug is considered highly permeable when the absolute BA is ≥ 85%. High permeability can also be concluded if ≥ 85% of the administered dose is recovered in urine as unchanged (parent drug), or as the sum of the parent drug, Phase 1 oxidative and Phase 2 conjugative metabolites.
Section 2 “Solid dispersion”
6. Solid dispersions are not a manufacturing method (going by the current title), it is an approach/technology and we can utilize e.g. quench cooling, spray drying, freeze drying as manufacturing methods to produce solid dispersions. So please think about an appropriate title for this manuscript.
The reviewer’s point is well appreciated
In the response of reviewer’s comment, the title was changed to “Overview of the Manufacturing Methods of Solid Dispersion Technology for Improving the Solubility of Poorly Water-Soluble Drugs and Application to Anticancer Drugs”.
7. I am sorry, but do not understand your “classification” described in 2.1 and also Figure 2 is not clear at all. Also, how this classification relates to anticancer drugs?
In the section 2.1 and Figure 2, we simply classified solid dispersion on the basis of the carriers used in the preparation of the formulation. It does not relate to anticancer drugs.
8. Again, Section 2.2 lacks examples of anticancer drugs.
In the section 2.2, we also simply classified solid dispersion on the basis of structure, it is not related to anticancer drugs.
9. Sections 2.3 and 2.4 should include advantages and disadvantages specific to anticancer drugs. At the moment these two sections are not informative and bring nothing new to this paper.
The reviewer’s point is very well appreciated.
In the response of reviewer’s comment, we added more advantage and disadvantage specific to anticancer drugs in section 2.3 and 2.4 as follows:
Line 252~253: Section 2.3:
Besides, SD technology showed advantage for anticancer drugs because of simple and cheaper in cancer treatment compared with surgery and radiotherapy.
Line 263~265: Section 2.4:
Patients suffering from cancer should continue to use anticancer drugs during treatment. However, the instability of SD during the period of storage can affect drug quality and the effectiveness of treatment.
10. Table I is redundant unless is contains anticancer molecules, but those are already presented in Table II.
Solid dispersion technology is widely applied for enhancing the solubility of poorly water-soluble drugs including anticancer drugs. So, in Table 1, we would like to list the manufacturing methods and drug of solid dispersion technology. And in the Table 2, we focused on the anticancer drugs.
11. For Section 2.5 there need to be a summary section as which of the method(s) is/are better for making solid dispersions of anticancer drugs. It needs to include aspects of health and safety, exposure to those molecules during production, routes of delivery, formulation etc. Also – it is of a key importance to highlight the differences in processing APIs versus biologics.
The reviewer’s point is very well appreciated.
In section 2.5, we added one more section as follows:
2.5.12. Suitable Methods for Making SD of Anticancer Drugs
Anticancer drugs were divided on the basis of biologic oncology drugs (monoclonal antibodies) and small-molecule-targeted oncology drugs (nonbiologic targeted oncology drugs) with a difference in their effectiveness and safety profiles. Biologic drugs are large proteins that are administered by IV injection while small molecule drugs are given orally.
Oral administration is currently preferred for treatment of cancer in comparison with IV route because it is convenient, painless, safe, and economic. Oral drugs can be administered at home, do not induce the same discomfort as an IV infusion, and the drug concentration can be maintained for long time periods in cancerous cells. In addition, oral dosage forms are easy to store and transport. Therefore, oral administration has received increasing attention, leading to increased numbers of anticancer drugs being developed for oral dosing.
To date, SD technology is widely used to improve the solubility and BA of anticancer drugs due to its simplicity, economy, and high effectiveness. Most of methods are suitable for making SD, in which melting method, solvent evaporation method, SCF technology, and freeze-drying are the better methods for making SD formulation of anticancer drugs in comparison with other methods. The selected method will be based on physicochemical properties of anticancer drugs.
Reviewer 2 Report
The manuscript entitled „Overview of the Manufacturing Methods of Solid Dispersion for Improving the Solubility of Poorly Water-Soluble Drugs” deals with the enhancement of water solubility of APIs by means of solid dispersion preparation. This review shows a classification of SDs and methods of preparation of thereof. It is also stats that the review is focused on SDs containing anticancer drugs.
In my opinion the added value of the manuscript is low; the methods and classification of SDs (partial, two groups are missing) is presented as in typical way, authors do not provide any critical consideration, new point of view). The issue of SDs with anticancer drugs is described only a little while in the introductory part the interest in this group of substance is emphasized. Moreover, section 2.6 contains many repetitions of information given in previous parts of the manuscript). The manuscript contains several logic errors, such as:
- Lines 20-21: something is missing in this sentence.
- Line 137: If the 1st generation is stable then why the 2nd was introduced?
- Line 182: SD is a mixture of API and a carrier. Did you mean that the drug is miscible with the carrier?
Moreover, several issues need further explanation:
- Line 205, 209: Not every SD is amorphous. What about the physical stability? Amorphous drugs can easily recrystallize upon storage (the effect of temperature and humidity). You should consider this effect.
- Line 214: mostly physical (chemical can occur mostly during processing, eg. The decomposition upon heating, however the physical stability seems to be the most crucial issue).
It would be beneficial if you try to group the methods presented in Figure 5. Otherwise it is a bit chaotic.
Given the aforementioned remark I would like to advise to reject the paper.
Author Response
- Lines 20-21: something is missing in this sentence.
In response to reviewer’s comment, we rewrite lines 20-21 as follows:
New line 22~25: “Oral administration is the preferred route for drug delivery due to several advantages such as low cost, pain avoidance, safety, and so on. The main problem of NCEs is a limited aqueous solubility, resulting in poor absorption and low bioavailability.”
- Line 137: If the 1st generation is stable then why the 2nd was introduced?
In the section 2.1, we classified the solid dispersion on the basis of carrier (Fig. 2). In the 1st generation, crystalline carriers were used. However, these carriers have the disadvantage of forming crystalline solid dispersions, which were more thermodynamically stable and did not release the drug as quickly as amorphous ones. Thus, 2nd generation was introduced.
- Line 182: SD is a mixture of API and a carrier. Did you mean that the drug is miscible with the carrier?
Yes, the drug is miscible with the carrier.
Moreover, several issues need further explanation:
- Line 205, 209: Not every SD is amorphous. What about the physical stability? Amorphous drugs can easily recrystallize upon storage (the effect of temperature and humidity). You should consider this effect.
The reviewer’s point is well appreciated.
We agreed with the reviewer, not every SD is amorphous. Practically, in SD formulation, to decrease the crystalline structure of drug in to amorphous form can help the formulation become more stable. The effect of temperature and humidity may be the reason of recrystallize of amorphous drugs. It is one of the disadvantages of SD that was presented in section 2.4. Therefore, we change “Drugs are in the amorphous state” to “To decrease the crystalline structure of drug in to amorphous form”.
- Line 214: mostly physical (chemical can occur mostly during processing, eg. The decomposition upon heating, however the physical stability seems to be the most crucial issue).
We changed “Chemical instability” to “Instability”
It would be beneficial if you try to group the methods presented in Figure 5. Otherwise it is a bit chaotic.
We would like to keep Fig. 5 as the original.
Reviewer 3 Report
Please see attached file

Author Response
I just wonder why the introduction is only focused on cancer drugs. Basically, all drug candidates that are under development to treat all diseases are mostly insoluble. Therefore, I think it is not necessary to put special attention to only cancer drugs since insolubility is applicable to all drugs. I advise the authors to revise this part.
The reviewer’s point is well appreciated.
In this review, we would like to focus on the manufacturing methods of solid dispersion technology for improving the solubility and bioavailability of poorly water-soluble drugs and then application of these methods for anticancer drugs. Therefore, the title will be changed to “Overview of the Manufacturing Methods of Solid Dispersion Technology for Improving the Solubility of Poorly Water-Soluble Drugs and Application to Anticancer Drugs”.
Another concept about insoluble drug is the dose/strength of the final product. There are many examples that the drugs are poorly soluble, but the dose is so small, ie <0.5/1 mg. In such case, the drug does not require any efforts to increase solubility. Author should add explanation about the dose/strength of the drug, elaborate it with its low solubility, and correlate it with the necessity of solid dispersion.
The reviewer’s point is well appreciated
In response to reviewer’s comment, we added explanation about the dose/strength of the drug, elaborate it with its low solubility, and correlate it with the necessity of solid dispersion as follows:
Line 111~123: In clinical, there are many insoluble drugs with small dose administration such as risperidone (0.25~4 mg), lorazepam (0.5~2 mg), diazepam (2~10 mg), clonazepam (0.5~2 mg), etc. that does not require to increase solubility for drugs. However, the solubility of these drugs is usually affected by pH due to physicochemical properties resulting in the decrease in the effective treatment. For example, risperidone is indicated for the treatment of schizophrenia with a small dose (0.25 mg, 0.5 mg, 1 mg, 2 mg, 3 mg, and 4 mg) in oral administration. It is a weak base that is practically insoluble in water. Its solubility is pH dependent with highly soluble in acidic pH, the solubility decreases as pH increases (range from >200 mg/mL at pH 2.1 down to 0.29 mg/mL at pH 7.6 and reaches 0.08 mg/mL at pH 8). After oral administration, risperidone is rapidly absorbed and approximately 80% of drugs will be absorbed in the GI tract, where the solubility significantly drops. Moreover, risperidone is a metabolized drug, in which approximately 70% and 14% of the dose is excreted in urine and feces, respectively. So, enhancing solubility in simulated intestinal pH to ensure higher drug concentrations at the main absorption site and improve BA is a challenge in drug development.
What is the main advantage of solid dispersion compared to salt formation, cocrystallization, complexation, micro-/nanonization? Authors should add comment related this matter to make it clear why solid dispersion is still being used and remains important in the pharmaceutical field. Authors can start from the challenge of salt formation or cocrystallization, ie not all of the drugs can ionize with all cation/anion, phase dissociation or stability issue is inherited in salt formation or cocrystallization, etc.
The reviewer’s point is well appreciated.
In response to reviewer’s comment, we added more advantages of solid dispersion technology compared to salt formulation, cocrystallization, and so on as follows:
Line 241~251, section 2.3: • In addition, SD showed an advantage compared to the salt formulation, cocrystallization, and so on. For example, salt formulations are an ionized molecule active pharmaceutical ingredients (APIs) (cationic or anionic form) and widely used in the pharmaceutical industry due to the broad capacity to design an API according to desired drug properties. However, not all of the drugs can ionize with all cation/anion, phase dissociation or stability issue is inherited in salt formation or cocrystallization, etc. Salt formulation showed several disadvantages such as reduce solubility, dissolution rate and thereby decrease relative BA (common ion effect for HCl salts), greater regulatory scrutiny for strong acid salt isolated from alkyl alcohols, increased hygroscopicity e.g. Na, K salts, spray drying/lyophilization can dissociate strong acid salts, etc. The disadvantages of salt formulation can resolve when the formulation was made by SD.
Before going to classification of solid dispersion, it is very important that the authors raise the definition of solid dispersion itself.
The definition of solid dispersion was added as follows:
Line 141~143: “SD is defined as a group of solid products consisting of a hydrophobic drug dispersed in at least one hydrophilic carrier, resulting in increased surface area, enhanced drug solubility and dissolution rate. It was classified as follows:”
I do understand that that the authors tried to include as many as possible the examples of solid dispersion. I would recommend including commercially available drug that is formulated as solid dispersion. Some comments, ie manufacturer, therapeutic use of the drug would be very helpful.
The reviewer’s point is well appreciated.
In response to reviewer’s comment, commercial solid dispersion table was added in the manuscript.
Page 9
Table II. List of commercial solid dispersions.
Products | Drugs | Polymers | Company |
Afeditab® | Nifedipine | Poloxamer or PVP | Elan Corp, Ireland |
Cesamet® | Nabilone | PVP | Lilly, USA |
Cesamet® | Nabilone | PVP | Valeant Pharmaceuticals, Canada |
Certican® | Everolimus | HPMC | Novartis, Switzweland |
Gris-PEG® | Griseofulvin | PEG | Novartis, Switzweland |
Gris-PEG® | Griseofulvin | PVP | VIP Pharma, Denmark |
Fenoglide® | Fenofibrate | PEG | LifeCycle Pharma, Denmark |
Nivadil® | Nivaldipine | HPC/HPMC | Fujisawa Pharmaceuticals Co., Ltd |
Nimotop® | Nimodipine | PEG | Bayer |
Torcetrapib® | Torcetrapib | HPMC AS | Pfizer, USA |
Ibuprofen® | Ibuprofen | Various | Soliqs, Germany |
Incivek® | Telaprevir | HPMC AS | Vertex |
Sporanox® | Itraconazole | HPMC | Janssen Pharmaceutica, Belgium |
Onmel® | Itraconazole | HPMC | Stiefel |
Prograf® | Tacrolimus | HPMC | Fujisawa Pharmaceuticals Co., Ltd |
Cymbalta® | Duloxetine | HPMC AS | Lilly, USA |
Noxafil® | Posaconazole | HPMC AS | Merck |
LCP-Tacro® | Tacrolimus | HPMC | LifeCycle Pharma, Denmark |
Intelence® | Etravirine | HPMC | Tibotec, Yardley, PA |
Incivo® | Etravirine | HPMC | Janssen Pharmaceutica, Belgium |
Rezulin® | Troglitazone | PVP | Pfizer, USA |
Isoptin SRE-240® | Verapamil | Various | Soliqs, Germany |
Isoptin SR-E® | Verapamil | HPC/HPMC | Abbott Laboratories, USA |
Crestor® | Rosuvastatin | HPMC | AstraZeneca |
Zelboraf® | Vemurafenib | HPMC AS | Roche |
Zortress® | Everolimus | HPMC | Novartis, Switzweland |
Kalydeco® | Ivacaflor | HPMC AS | Vertex |
Kaletra® | Lopinavir and Ritonavir | PVP/polyvinyl acetate | Abbott Laboratories, USA |
I don’t understand why Table 2 is only focused on anti-cancer drugs. The title of this review is very general, but the introduction and study cases presented here are so specific to anti-cancer drug.
In this review, we would like to review the manufacturing methods of solid dispersion technology for improving the solubility and BA of poorly water-soluble dugs and application to anticancer drugs, therefore, in Table I we list of drugs investigated for solid dispersion and Table II (changed to Table III) we only focused on anticancer drugs. We also changed the title to “Overview of the Manufacturing Methods of Solid Dispersion Technology for Improving the Solubility of Poorly Water-Soluble Drugs and Application to Anticancer Drugs”.
As this review is focused on manufacturing methods. I am very keen to see the manufacturing development of solid dispersion from the lab scale, to pilot scale, and lastly to manufacturing scale. Not all method is available for commercial supply. I believe this will increase readership to this review.
The reviewer’s point is well appreciated.
In response to reviewer’s comment, we added more information relates to manufacturing development of solid dispersion as follows:
2.5.13. Lab Scale and Industrial Scale of Manufacturing Processes
Several manufacturing methods were used to making SD. However, not all method is available for commercial process. Practically, the melting method and solvent evaporation method are two distinct processes that are widely used in the lab scale and industrial scale.
In the lab scale, for the solvent evaporation method, rotary evaporator was mostly used to produce SD. Recently, SCF and freeze-drying are also employed. Melting method its simplicity and economy are popularly used. Currently, several types of equipment from many manufacturers such as Brabender Technologies, Coperion GmbH, Thermo Fisher Scientific, Leistritz Advanced Technologies Corp etc. are available in the laboratory, in which, SD amount can be produced from few grams to a kilogram.
In the industrial scale, production of SD is not simple as lab scale because of a large amount of product from a few kilograms to several hundred kilograms. Besides, processes need to be robust, reproducible, and it has to follow good manufacturing practices (GMP). Therefore, it is hard to ensure for processes such as solvent cast evaporation or water bath melting process. Spray-drying and freeze-drying are the most representative of the solvent evaporation methods used in the industry for manufacturing SD. Moreover, the spray-drying process is easy to scale up from lab scale to industrial scale. Melt agglomeration and hot-melt extrusion are two types of melting process that is available on the industrial scale. For instance, hot-melt extrusion is one of the most method used in an industrial scale to produce SD using twin-screw extruder with a large diameter of the screw (16~50 mm) compared with small diameter of the screw in lab scale (11~16 mm).
In summary, the selected method for manufacturing process plays an important role in the success of formulation. In the lab scale, the criteria for selecting the melting method based on the melting point and thermal stability. For selecting the solvent evaporation method, an important factor to consider are properties of the drug, carrier, and an organic solvent. In the industrial scale, the production of SD is limited to only a few manufacturing processes. Hot-melt extrusion is the most common method among the melting process to produce SD. For the evaporation method, the selection criteria are based on solvent toxicity and loading capacity.
It would be very helpful if there are some short comments of basic principle of each preparation methods for solid dispersion as this review is focused on manufacturing process.
The reviewer’s point is very well appreciated
In response to reviewer’s comment, we added more basic principle of melting method, solvent evaporation method, and supercritical fluid (SCF) as follows:
Line 273~276: Melting method: The basic principle of melting method is that a physical mixture of a drug and hydrophilic carrier is heated directly until they melt at a temperature slightly above their eutectic point. Then, the melt is cooled and solidified rapidly in an ice bath with stirring. The final solid mass is crushed and sieved.
Line 293~298: Solvent evaporation method: This method was developed mainly for heat unstable components because drug and carrier are mixed by a solvent instead of heat as in melting method. Therefore, this method allows to use carriers with a high melting point which is too high for melting method. The basic principle of this method is that drug and carrier are dissolved in a volatile solvent in order to mix them homogeneously. SD is obtained by evaporating the solvent under constant agitation. Then, the solid SD is crushed and sieved.
Line 407~410: Supercritical fluid (SCF): The basic principle of SCF is that the drug and carrier are dissolved in a supercritical solvent (e.g. CO2) and sprayed through a nozzle into an expansion vessel with lower pressure. The rapid expansion induces rapid nucleation of the dissolved drugs and carriers, leading to the formation of SD particles with a desirable size distribution in a very short time.
The other basic principles of remaining methods (melting evaporation, melt agglomeration process, hot-melt extrusion method, lyophilization, electrospinning method, co-precipitation, spray drying, and kneading method) have been showed in the manuscript as follows:
Line 327~330: Melting evaporation method: The melting solvent method combines melting method and solvent evaporation method. The drug is first dissolved in a suitable solvent and incorporated into the melt of the carrier, and the mixture is then evaporated to complete dryness. Practically, this method is very useful for drugs with a high melting point.
Line 336~338: Melt agglomeration process: Melt agglomeration is a process in which a binder acts as a carrier. In this method, the drug, binder, and other excipients are heated to a temperature above the melting point of the binder. Alternatively, a dispersion of the drug is sprayed onto the heated binder.
Line 347~352: Hot-melt extrusion method is a common method for improving solubility and oral BA of poorly water-soluble drugs, in which the amorphous SD was formed without solvent, thereby avoiding residual solvents in the formulation [113]. This method is conducted by a combination of the melting method and an extruder, in which a homogeneous mixture of drug, polymer, and plasticizer is melted and then extruded through the equipment. The shape of products at the outlet of extruder can be controlled and do not need to grind in the final step.
Line 367~370: Lyophilization Techniques/Freeze Drying: Lyophilization is an alternative process to the solvent evaporation method in which the drug and carrier are dissolved in a solvent and then the solution is frozen in liquid nitrogen to form a lyophilized molecular dispersion [116]. This method is typically used for thermolabile products that are unstable in aqueous solutions but stable in the dry state for prolonged storage periods.
Line 384~387: Electrospinning method: In this method, solid fibers are produced from a polymeric fluid stream or melt delivered through a millimeter-scale nozzle [119]. The advantage of this method is that the process is simple and inexpensive. This method is suitable for preparing nanofibers and controlling release of biomedicine.
Line 394~397: Co-precipitation: In this method, the carrier is first dissolved in solvent to prepare a solution and the drug is incorporated into the solution with stirring to form a homogeneous mixture. Then, water is added dropwise to the homogenous mixture to induce precipitation. Finally, the precipitate is filtered and dried.
Line 430~435: Spray drying: In this method, the drug is dissolved in a suitable solvent and the carrier is dissolved in water in order to prepare the feed solution. Then, the two solutions are mixed by sonication or other suitable methods until the solution is clear. In the procedure, the feed solutions were firstly sprayed to drying chamber via a high pressure nozzle to form fine droplets. The formed droplets in drying chamber are made up with the drying fluid (hot gas) to form particles under nano or micro size.
Line 446~448: Kneading method: In this method, the carrier is dispersed in water and processed into a paste. Then, the drug is added and kneaded thoroughly. The final kneaded formulation is dried and passed through a sieve if necessary.
It will be good to add perspective/future prospect section in this review.
The reviewer’s point is well appreciated.
In response to reviewer’s comment, we added “Future prospect” section in the manuscript as follows:
2.7 Future prospects
SD is currently considered one of the most effective methods for enhancing the solubility and BA of poorly water-soluble drugs. Even though the issues related to preparation, stability, and storage formulation of drugs may be limited the numbers of commercial SD products on the market. However, by improving manufacturing methods and carriers to solve these problems, SD product is still steadily increasing in clinical.
In recent years, carriers were used in the preparation of SD with many developments. Some studies used new carriers while other studies used more than one carrier for making SD formulation. By using more than one carrier in the preparation of SD, the effectiveness of the method can increase, the recrystallization decrease and the stability of SD can improve. Some carriers used in recently are Inulin®, Gelucire®, Pluronic®, and Soluplus®. In the manufacturing process, Kinetisol Dispersing (KSD) is a novel high energy mixing process for the preparation of SD, in which the drug and carrier were processed by utilizing a series of rapidly rotating blades through a combination of kinetic and thermal energy without the aid of external heating sources. This brings new hope to develop more SD products in the future.
Round 2
Reviewer 1 Report
This version of the manuscript has been much improved however, a number of issues still need to be addressed:
1. Despite an earlier comment in relation to improving English this was not completed. Please give this manuscript to a native speaker.
2. The use of word “classification” for solid dispersions is confusing. Normally we would use the word “class” or “type”. Please see comment #1.
3. Page 7, line 234 “To decrease the crystalline structure of drug into amorphous form” – this sentence does not make any sense. Please see comment #1.
4. Page 7, line 252 “Besides, SD technology showed advantage for anticancer drugs because of simple and cheaper in cancer treatment compared with surgery and radiotherapy.” – this is a very dangerous and wrong statement. The selection of the best cancer therapy depends on the specialist medical team and not because SD might have the potential to replace surgery and radiotherapy. This is either poor choice of words or oblivion. Please see comment #1.
5. The authors in rebuttal state that in Table 2 the focus was anticancer drugs. Is e.g. nifedipine an anticancer drug? Ibuprofen??? Authors really should put some efforts into presenting correct and factual information.
6. References missing in the “Future prospects” section in relation to the Kinetisol technology.
Author Response
1. Despite an earlier comment in relation to improving English this was not completed. Please give this manuscript to a native speaker.
English has been improved as reviewer’s comment. Please refer the attached file.
2. The use of word “classification” for solid dispersions is confusing. Normally we would use the word “class” or “type”. Please see comment #1.
In response to reviewer’s comment, the word “classification” was changed to “class”.
3. Page 7, line 234 “To decrease the crystalline structure of drug into amorphous form” – this sentence does not make any sense. Please see comment #1.
The reviewer’s point is very well appreciated.
The sentence “To decrease the crystalline structure of drug into amorphous form” was removed from the revised manuscript.
4. Page 7, line 252 “Besides, SD technology showed advantage for anticancer drugs because of simple and cheaper in cancer treatment compared with surgery and radiotherapy.” – this is a very dangerous and wrong statement. The selection of the best cancer therapy depends on the specialist medical team and not because SD might have the potential to replace surgery and radiotherapy. This is either poor choice of words or oblivion. Please see comment #1.
Thank you for comments. In response to reviewer’s comment, the sentence was rewrite as follows:
Page 7, line 249-254: “Practically, dissolution of drugs is a prerequisite for complete absorption to have the desired therapeutic effect of anticancer drugs after oral administration. Most of the anticancer drugs exhibit poor aqueous solubility causes of dissolution limit resulting low BA and high variability in blood concentration. The limitation of drug dissolution can improve by SD, a technique that induces supersaturated drug dissolution and with that it enhances in vivo absorption”.
5. The authors in rebuttal state that in Table 2 the focus was anticancer drugs. Is e.g. nifedipine an anticancer drug? Ibuprofen??? Authors really should put some efforts into presenting correct and factual information.
Originally, we listed Table II focus on anticancer drugs and then, in the new version of the manuscript, we added one more Table about “list of commercial solid dispersions” (new Table II), Table II (original version) was named as Table III (new version). So, in new Table II, we just listed the commercial solid dispersions, not focused on anticancer drugs.
6. References missing in the “Future prospects” section in relation to the Kinetisol technology.
In response to reviewer’s comment, references of the Kinetisol technology were added as follows:
171. Bennett, R.C.; Brough, C.; Miller, D.A.; O’Donnell, K.P.; Keen, J.M.; Hughey, J.R.; Williams, R.O.; McGinity, J.W. Preparation of amorphous solid dispersions by rotary evaporation and KinetiSol dispersing: approaches to enhance solubility of a poorly water-soluble gum extract. Drug Dev. Ind. Pharm. 2015, 41, 382–397.
172. Keen, J.M.; LaFountaine, J.S.; Hughey, J.R.; Miller, D.A.; McGinity, J.W. Development of itraconazole tablets containing viscous KinetiSol solid dispersions: In vitro and in vivo analysis in dogs. AAPS PharmSciTech 2018, 19, 1998–2008.
173. DiNunzio, J.C.; Brough, C.; Miller, D.A.; Williams, R.O.; McGinity, J.W. Fusion processing of itraconazole solid dispersions by Kinetisol® dispersing: A comparative study to hot melt extrusion. J. Pharm. Sci. 2010, 99, 1239–1253.
Reviewer 2 Report
Dear Authors,
The corrections you included make the manuscript much better than it was in its former version. However, the text still suffers from several language errors. I suggest to ask an English native speaker for a correction.
Author Response
Lines 20-21: something is missing in this sentence.
Original line 20-21: “Thus, oral administration is the preferred route for drug delivery. Because most of NCEs are poor aqueous solubility resulting in poor absorption and poor bioavailability. Thus, improving oral bioavailability of poorly water-soluble drugs is a great challenge in the pharmaceutical industry”.
As the reviewer pointed out, we rewrote the sentence as follows:
Line 22-25: “Oral administration is the preferred route for drug delivery due to several advantages such as low cost, pain avoidance, and safety. The main problem of NCEs is a limited aqueous solubility, resulting in poor absorption and low bioavailability. Thus, improving oral bioavailability of poorly water-soluble drugs is a great challenge in the development of pharmaceutical dosage forms”.
Line 137: If the 1st generation is stable then why the 2nd was introduced?
Thank you for comments. Here, there is a typo error in this line. We edited the sentence in line 137 as follows:
In revised manuscript line 168-169: Because of thermodynamic instability of first class SD [44], second class SDs were introduced using amorphous polymeric carriers [45] instead of urea or sugars.
Line 182: SD is a mixture of API and a carrier. Did you mean that the drug is miscible with the carrier?
Thank you for comments. Herein, SD is a mixture of API and carrier. Therefore, the line 182 was changed to:
In revised manuscript line 212: Herein, SD is a mixture of the drug and a carrier.
Moreover, several issues need further explanation:
Line 205, 209: Not every SD is amorphous. What about the physical stability? Amorphous drugs can easily recrystallize upon storage (the effect of temperature and humidity). You should consider this effect.
We agreed with the reviewer, not every SD is amorphous. In advantage section (2.3), we rewrote and added more about the advantage of SD (Line 231~254).
As the reviewer’s comment, amorphous drugs can easily recrystallize upon storage (the effect of temperature and humidity). It is right and it was discussed in section 2.4 line 260~264 as follows:
“Due to their thermodynamic instability, SD is sensitive to temperature and humidity during storage. These factors can promote phase separation and crystallization of SD by increasing the overall molecular mobility, decreasing the glass transition temperature (Tg) or disrupting interactions between the drug and carrier, resulting in a decreased solubility and dissolution rate of the drug”.
Line 214: mostly physical (chemical can occur mostly during processing, eg. The decomposition upon heating, however the physical stability seems to be the most crucial issue).
Thank you for comments. According to the reviewer’s suggestion, “Chemical instability” (Line 214-original) was changed to “Physical instability” (Line 258 revised manuscript).
It would be beneficial if you try to group the methods presented in Figure 5. Otherwise it is a bit chaotic.
As the reviewer suggested, the method in Fig. 5 was grouped as follows:
Reviewer 3 Report
Dear Editor,
I would like to propose this manuscript being published in your journal after minor English editing that can be done during proof reading preparation.
Author Response
We revised the manuscript as to the reviewer's comments.
Round 3
Reviewer 1 Report
Thank you for all the changes. No more comments.